# Hygroscopic behavior of atmospheric aerosols containing nitrate salts and water-soluble organic acids

Bo Jing[1], Zhen Wang[1, 2], Fang Tan[1, 2], Yucong Guo[1], Shengrui Tong[1], Weigang Wang[1], Yunhong Zhang[3], and Maofa Ge[1, 2, 4]

[1]State Key Laboratory for Structural Chemistry of Unstable and Stable Species, CAS Research/Education Center for Excellence in Molecular Sciences, Institute of Chemistry, Chinese Academy of Sciences, Beijing 100190, P. R. China

[2]University of Chinese Academy of Sciences, Beijing 100049, P. R. China

[3]The Institute of Chemical Physics, School of Chemistry and Chemical Engineering, Beijing Institute of Technology, Beijing 100081, P. R. China

[4]Center for Excellence in Regional Atmospheric Environment, Institute of Urban Environment, Chinese Academy of Sciences, Xiamen 361021, P. R. China

*Correspondence to:* Maofa Ge (gemaofa@iccas.ac.cn)

**Abstract**

While nitrate salts have critical impacts on environmental effects of atmospheric aerosols, the effects of coexisting species on hygroscopicity of nitrate salts remain uncertain. The hygroscopic behaviors of nitrate salt aerosols ($NH_4NO_3$, $NaNO_3$, $Ca(NO_3)_2$) and their internal mixtures with water soluble organic acids were determined using a hygroscopicity tandem differential mobility analyzer (HTDMA). The nitrate salt/organic acid mixed aerosols exhibit varying phase behavior and hygroscopic growth depending upon the type of components in the particles. Whereas pure nitrate salt particles show continuous water uptake with increasing RH, the deliquescence transition is still observed for ammonium nitrate particles internally mixed with organic acids such as oxalic acid and succinic acid with a high deliquescence point. The hygroscopicity of submicron aerosols containing sodium nitrate and an organic acid is also characterized by continuous growth, indicating that sodium nitrate tends to exist in a liquid-like state under dry conditions. It is observed that in contrast to the pure components the water uptake is hindered at low and moderate RH for calcium nitrate particles containing malonic acid or phthalic acid, suggesting the potential effects of mass transfer limitation in highly viscous mixed systems. Our findings improve fundamental understanding of the phase behavior and water uptake of nitrate salt-containing aerosols in the atmospheric environment.

## 1 Introduction

Atmospheric aerosols exert significant impacts on the earth's radiation balance by absorbing or scattering solar radiation and modifying the properties of clouds, which result in large uncertainty in climate forcing (Haywood and Boucher, 2000; Carslaw et al., 2013). The hygroscopic particles can provide liquid water medium for multiphase and aqueous phase chemical processes that influence chemical constituents in the condensed and gas phase (Wang et al., 2016a; Cheng et al., 2016; Hodas et al., 2014; Faust et al., 2017; Tan et al., 2016; Li et al., 2017). It has been established that aerosol liquid water could promote the formation of secondary organic aerosol (SOA) through the partitioning of gas-phase water-soluble organic compounds to the condensed phase and subsequent aqueous-phase processing (Hodas et al., 2014; Faust et al., 2017). The hygroscopic behavior and water content of aerosols in the atmosphere are highly dependent upon the chemical composition, mixing state and ambient relative humidity (RH) (Martin, 2000; Choi and Chan, 2002; Krieger et al., 2012). Atmospheric process or chemical aging plays an important role in the chemical composition, mixing state of aerosol particles and thus impacts aerosol hygroscopicity (Boreddy et al., 2015; Boreddy et al., 2014b). The atmospheric particulate matter from multiple biogenic and anthropogenic sources is commonly composed of complex inorganic and organic compounds with various physicochemical properties.

The nitrate salts are ubiquitous and account for a large fraction of inorganic constituents within the atmospheric particulate matter in urban/polluted environments, especially in winter (Huang et al., 2014; Zhang et al., 2015). For example, chemical analyses have shown that nitrate typically constitutes a fraction (7−14%) of the total particulate matter during the high pollution events at the urban sites of China (Huang et al., 2014). The nitrate content can even dominate 22%-24% of mass fractions of particulate matter in some urban areas such as Beijing and Los Angeles (Zhang et al., 2015). The majority of nitrate salts in ambient particles exists as $NH_4NO_3$, $NaNO_3$ and $Ca(NO_3)_2$ depending on the environmental conditions and chemical formation mechanisms. The formation of nitrate salts is typically attributed to atmospheric reactions of ammonia, sea salt and mineral dust with nitric acid or nitrogen oxides such as $NO_2$, $NO_3$ and $N_2O_5$ (Zhang et al., 2015). In the urban area, due to the considerable influence of anthropogenic sources the major chemical form of nitrate salts in fine particulate matter is ammonium nitrate generated via the heterogeneous reaction between $HNO_3$ and $NH_3$ in the aerosol phase. Field measurements and laboratory studies have indicated that the mineral dust ($CaCO_3$) and sea salt (NaCl) emitted from natural sources could undergo atmospheric aging through the heterogeneous reactions with nitric acid or nitrogen oxides, resulting in the formation of $Ca(NO_3)_2$ and $NaNO_3$ (Zhang et al., 2015). The inorganic salts in the particle phase are generally internally mixed with organic compounds that contribute a large fraction of fine particulate matter. Field measurements have confirmed that organic fraction of the aerosols contains large amounts of water-soluble organic compounds (WSOCs) (Saxena

and Hildemann, 1996; Gysel et al., 2004; Decesari et al., 2005), which affect the hygroscopicity of inorganic components (Boreddy et al., 2014a; Boreddy and Kawamura, 2016). It has been found that water-soluble organic acids such as dicarboxylic acids are representative and important constituents of the WSOCs (Kawamura and Ikushima, 1993; Kawamura et al., 1996; Decesari et al., 2006; Kawamura and Bikkina, 2016). Among the water-soluble organic acids, oxalic acid, malonic acid, succinic acid and phthalic acid are typically identified as the most abundant dicarboxylic acids in atmospheric particles (Kawamura and Bikkina, 2016).

The phase state of aerosol particles is an important factor in determining the particle's physicochemical properties and climate effects (Kanakidou et al., 2005). Due to its significance, the comprehensive understanding of particle phase in aerosols with varying components could improve the prediction ability of climate models (Kanakidou et al., 2005; Virtanen et al., 2010; Shiraiwa et al., 2017). Unlike sulfates such as ammonium sulfate with distinct phase transition behaviors, the nitrate salts including ammonium nitrate especially in submicron particles tend to take up water continuously from low RH without obvious phase transitions (Mikhailov et al., 2004; Gibson et al., 2006). Previous observations indicated that pure $NH_4NO_3$ droplets were difficult to homogeneously crystallize and thus remained in the liquid-like state even under extremely dry conditions (<1% RH) while it could efflorescence with the presence of solid core in the droplets (Lightstone et al., 2000). Soluble inorganic crystalline particles acting as the contact nuclei were found to induce crystallization of aqueous ammonium nitrate (Davis et al., 2015). It was found that the phase state of $NH_4NO_3$ tended to be substantially affected by coexisting species such as ammonium sulfate and succinic acid (Lightstone et al., 2000; Liu et al., 2016). The previous hygroscopic studies have focused on the hygroscopicity of ammonium sulfate and its relevant mixtures with organics such as water-soluble organic acids (Cruz and Pandis, 2000; Choi and Chan, 2002; Prenni et al., 2003; Wise et al., 2003; Badger et al., 2006; Hodas et al., 2015; Wang et al., 2017). However, the overall role of the water-soluble organic acids in hygroscopic growth and phase behavior of the nitrate salts remains uncertain. Thus, related studies are of importance for understanding of their environmental effects.

In this work, the hygroscopic behaviors of internally mixed aerosols composed of atmospherically relevant nitrate salts and water-soluble organic acids are determined under subsaturation conditions with a hygroscopicity tandem differential mobility analyzer (HTDMA) system. The measured hygroscopic growth of relevant aerosols is compared with predictions from the Zdanovskii-Stokes-Robinson (ZSR) method based on hygroscopicity of individual components. The significant effects of water-soluble organic acids with various hygroscopic characteristics on water uptake behaviors of nitrate salts have been confirmed, and relevant atmospheric implications are discussed.

**2 Experimental and method section**

## 2.1 Reagents

Table 1 summarizes chemical properties and manufacturer of nitrate salts (ammonium nitrate, sodium nitrate and calcium nitrate) and water-soluble organic acids (oxalic, malonic, succinic and phthalic acids) in this study. Aerosol particles were generated from a 0.1 wt% aqueous solution of pure component or mixtures containing nitrate salts and organic acids at a specific mass ratio. The corresponding solutions were prepared using ultrapure water (EASY Pure® II UF ultrapure water system, resistivity ≥ 18.2 MΩ cm).

## 2.2 Hygroscopic growth measurements

In this study, the hygroscopicity tandem differential mobility analyzer (HTDMA) system was used to measure the hygroscopic growth of aerosol particles studied. This system has been fully described in our previous studies (Jing et al., 2016; Peng et al., 2016; Jing et al., 2017). Briefly, the HTDMA setup consists of the aerosols' generation and drying section, humidity control apparatus and the particle size selection and detection system. The polydisperse particles were generated from a constant output atomizer containing sample solution, and then passed through a silica gel diffusion dryer combined with a Nafion gas dryer to be dried to RH < 5%. After charged by a neutralizer, the dry particles were transformed into quasi-monodisperse particles with mobility diameter of 100 nm by the first differential mobility analyzer (DMA1). The size-selected aerosols then entered the humidity control apparatus where they were exposed and equilibrated to a given RH in the range of < 5% to 90% with a residence time of 5 s. The size distribution of aerosols after humidification was determined by the second DMA (DMA2) and the condensation particle counter (CPC, MSP 1500). Sheath air in the DMA2 was drawn from the humidity control section to ensure no change in humidity of aerosol flow in the DMA2. The RH of sheath flow was measured at the outlet of DMA2 using a dew point hygrometer (Michell, UK) with an uncertainty of ± 0.8% RH and ± 0.1 K. The sheath and aerosol flow rates in both DMAs were 3.0 and 0.3 liters per minute, respectively. The inversion of HTDMA measurement data was based on a log-normal size distribution approximation (Stolzenburg and McMurry, 2008). All hygroscopicity measurements were conducted at ambient temperature (295 ± 1 K). During the experiment process, no obvious evaporation (no obvious decrease between dry particle size selected by DMA1 and that measured by DMA2) of nitrate salt -containing particles was observed under our measurement conditions.

The hygroscopic growth factor (GF) is calculated by: $GF = D_{wet}/D_{dry}$, where $D_{wet}$ is the diameter of particles measured by DMA2 at a particular RH and $D_{dry}$ is the diameter of dry particles measured by DMA2 at ≤ 5% RH. Considering the curvature effect for submicron droplets, the RH was converted to water activity $a_w$ in subsequent data analysis by the Köhler equation:

$$RH = a_w \exp(\frac{4M_w \sigma_{sol}}{RT \rho_w D_p})  \tag{1}$$

Here, $M_w$ is the molar mass of water, $\sigma_{sol}$ is the surface tension of the droplet, $R$ is ideal gas constant, $T$ is temperature, $\rho_w$ is the density of water, and $D_p$ is the droplet diameter. In this study, the effects of solutes on the surface tension of the droplet were not obvious and thus the surface tension of pure water (0.072 J m$^{-2}$) was used in estimates by Eq. (1). During the experimental period, hygroscopic growth of ammonium sulfate was measured regularly to validate the measurement reliability of the HTDMA. The RH measurement uncertainty was determined by measuring hydration curves of three inorganic salts such as sodium chloride (NaCl), ammonium sulfate [(NH$_4$)$_2$SO$_4$] and potassium chloride (KCl). It was found that the discrepancies between measured deliquescence points and theory predictions for the three salts were within $\pm$ 1.5% RH, which was used to estimate the measurement uncertainty in RH. All measured GFs at a given RH are the average values of at least three repeated measurements with the corresponding standard deviation less than 0.02. The measured deliquescence point and GFs of ammonium sulfate agreed well with our previous studies and other reported literature values (Peng et al., 2016; Jing et al., 2017; Sjogren et al., 2007). For example, the hygroscopic growth factors of 100 nm ammonium sulfate particles at 80% RH and 90% RH were determined to be 1.44 and 1.68, respectively, consistent with the literature results (GF(80% RH) = 1.46 and GF(90% RH) = 1.69, Sjogren et al., 2007).

The morphology of suspended particles was observed by scanning electron microscopy (SEM, S-4300, Hitachi). The SEM samples were prepared by using the aerosol generation, drying and humidification section of the HTDMA system without undergoing the size selection. The particles were deposited onto the silicon wafers, which were either put at the output of the silica gel diffusion dryer with RH less than 10%, or at the output of the Nafion humidification tube with 30% RH. The deposited particles were used for SEM measurement.

**2.3 Model methods**

The Aerosol Inorganic-Organic Mixtures Functional groups the Activity Coefficients (AIOMFAC) model is applied to estimate the hygroscopic growth of the three nitrate salts. The AIOMFAC has been developed to describe the activities of atmospherically relevant aqueous solutions up to high ionic concentrations at room temperature (Zuend et al., 2008; Zuend et al., 2011). The AIOMFAC predictions for nitrate salts are based on fitted water activity data. The fitted water activity data of ammonium nitrate are derived from measurements by Hamer and Wu (1972) and Robinson and Stokes (2002). The sodium nitrate data are based on measurements from Hamer and Wu (1972) and Tang and Munkelwitz (1994). The fitted water activity data of calcium nitrate are derived from measurements by Robinson and Stokes (2002).

The Zdanovskii-Stokes-Robinson (ZSR) method is used to estimate the hygroscopic growth of internally mixed particles by assuming that each component in the mixed particles takes up water independently (Malm

and Kreidenweis, 1997; Stokes and Robinson, 1966). The total water uptake by mixed particles is the sum of water content associated with each pure component, which can be expressed by the following equation:

$$GF_{mixed} = \left( \sum_k \varepsilon_k \cdot GF_k^3 \right)^{1/3} \tag{2}$$

, where $GF_{mixed}$ and $GF_k$ is the hygroscopic growth factor of the mixed particle and component $k$, respectively, and $\varepsilon_k$ is the volume fraction of component $k$ in the dry particle. The volume fraction $\varepsilon_k$ is obtained by:

$$\varepsilon_k = \frac{\left( w_k / \rho_k \right)}{\sum_i \left( w_i / \rho_i \right)} \tag{3}$$

Here $w_k$ is the mass fraction of component $k$, and $\rho_k$ is the density of component $k$. Although ZSR relation does not take into account the interactions between components in the mixture, this simple method has been found to be a valuable tool to predict the water uptake of atmospherically relevant mixtures under high RH conditions (Prenni et al., 2003). In the present study, the ZSR calculations are derived from the hydration curve of nitrate salts predicted from the AIOMFAC model and the three-parameter fit curve of organic acids (seen in Fig. S1, Supplement).

According to κ-Köhler theory proposed by Petters and Kreidenweis (2007), a single hygroscopicity parameter κ can be obtained based on hygroscopic growth measurements under subsaturated conditions (Carrico et al., 2008):

$$\kappa = \frac{\left( GF^3 - 1 \right)\left( 1 - a_w \right)}{a_w} \tag{4}$$

Where $a_w$ is the water activity and GF is the hygroscopic growth factor. Petters and Kreidenweis (2007) found that the κ-values derived from the hygroscopic growth data at RH~ 90% are generally consistent with that derived from the measured cloud condensation nucleus (CCN) activity data. The GFs of pure nitrate salt particles at 90% RH are converted into κ-values by using Eq. (4).

**3 Results and discussion**

**3.1 Water uptake behaviors of single components**

Table 2 summarizes hygroscopic growth factors of nitrate salt containing particles in this study. The hygroscopic growth of single components including three nitrate salts is shown in Fig. 1. The three nitrate salts present gradual hygroscopic growth without prompt deliquescence transitions over the whole RH range studied. The smooth hygroscopic growth suggests the nitrate salt particles likely remain in a liquid-like or amorphous state at the lowest RH, as also observed by previous studies (Lightstone et al., 2000; Gibson et al., 2006). For example, the early HTDMA studies by Mikhailov et al. (2004) and Svenningsson et al. (2006) have also

observed the continuous water-uptake behavior of submicron ammonium nitrate particles without any prompt deliquescence transition. Although crystalline sodium nitrate has a deliquescence point at 74.5% RH (Tang and Munkelwitz, 1994), the prior studies have found that micron and submicron $NaNO_3$ particles formed from aqueous solutions exhibited continuous hygroscopic growth due to initial particles in a metastable or amorphous form even at zero relative humidity (Gysel et al., 2002; Hoffman et al., 2004; Gibson et al., 2006). In this study, the measured growth factor for 100 nm $NaNO_3$ particles is 1.97 at 90% RH in good agreement with the literature value of 1.91 (Gysel et al., 2002). For calcium nitrate, $Ca(NO_3)_2$ particles gradually take up water with increasing RH and the measured hygroscopic growth generally matches the subsequent theory prediction based on the assumption that submicron $Ca(NO_3)_2$ particles at low RH exist as amorphous hydrates (tetrahydrate, $Ca(NO_3)_2 \cdot 4H_2O$) (Gibson et al., 2006). The GF of 100 nm $Ca(NO_3)_2$ particles is determined to be 1.56 and 1.89 at 80% and 90% RH, respectively, consistent with the corresponding predicted value of 1.52 and 1.82. As shown in Fig. S3, the nitrate salt particles stay in a liquid-like state at low RH less than 10%. The Raman spectrum of ammonium nitrate shows no obvious absorption between 3400 $cm^{-1}$ and 3600 $cm^{-1}$ for water peak feature, indicating the liquid-like particle appears to be amorphous solid. As shown in Fig. 1, the measured GFs of the three nitrate salts over the RH range studied are in fair agreement with the predictions from the AIOMFAC model based on the dehydration process, which also suggests nitrate salt particles remain noncrystalline even at the lowest RH. Compared to ammonium and sodium nitrates, $Ca(NO_3)_2$ can form amorphous hydrates under dry conditions (Gibson et al., 2006). On the basis of thermodynamic considerations, tetrahydrate ($Ca(NO_3)_2 \cdot 4H_2O$) is the stable hydrate for calcium nitrate. The previous study has also applied tetrahydrate data to calculate the theoretical growth factors of $Ca(NO_3)_2$ due to the lack of detailed amorphous hydrates data (Gibson et al., 2006). Similarly, the AIOMFAC predictions for $Ca(NO_3)_2$ in our study are based on the tetrahydrate data. It can be expected that the discrepancies between properties (density and molar mass) of amorphous hydrates and tetrahydrate could result in the slight deviation between AIOMFAC predictions and measurements. The κ-values of pure nitrate salt particles are derived from the GFs at 90% RH by using Eq. (4). The calculated κ-value in this study is 0.537, 0.821 and 0.658 for ammonium nitrate, sodium nitrate and calcium nitrate, respectively, in agreement with the corresponding value of 0.577 for ammonium nitrate and 0.80 for sodium nitrate in the literature (Petters and Kreidenweis, 2007).

For the organic acids studied (seen in Fig. S1, Supplement), oxalic acid (OA) and succinic acid (SA) present no deliquescence transition or any obvious water uptake below 90% RH, which agree well with previous studies (Jing et al., 2016; Peng et al., 2001). The observation of no evaporation losses (no decrease between dry particle size selected by DMA1 and that measured by DMA2) for dry OA particles is consistent with other HTDMA studies, suggesting that the generated particles after drying process exist as crystalline OA dihydrate

with low vapor pressure (Prenni et al., 2001; Mikhailov et al., 2009). The GFs of aqueous OA from the
dehydration curve measured by Mikhailov et al. (2009) are applied to estimate ZSR predictions when assuming
OA in the mixture contributes to water uptake at high RH. The malonic acid (MA) and phthalic acid (PA)
exhibit continuous water uptake behaviors across all RH studied, consistent with the observations in other
studies (Prenni et al., 2001; Brooks et al., 2004; Jing et al., 2016). The measured GF of 100 nm malonic acid
particles at 80% RH and 90% RH is 1.34 and 1.65, respectively, consistent with corresponding literature value
of 1.37 and 1.73 (Prenni et al., 2001). 100 nm phthalic acid particles have a GF of 1.14 at 80% RH, close to the
value of ~1.12 reported by Brooks et al. (2004).
**3.2 Water uptake behaviors of $NH_4NO_3$/organic acid mixtures**
The hygroscopic behaviors of 100 nm $NH_4NO_3$ particles internally mixed with organic acids at different mass
ratios are shown in Figs. 2 and 3. As shown in Fig. 2, the hygroscopic growth of $NH_4NO_3$/oxalic acid (OA)
mixed particles shows various features with varying oxalic acid content. For the 3:1 $NH_4NO_3$/OA mixed system,
the GFs of particles increase gradually with elevated RH and no prompt deliquescence behavior is observed in
the RH range studied. The ZSR rule could well describe the hygroscopic growth of mixed particles when taking
into account the dissolution of oxalic acid at RH higher than 70%, as indicated by the good agreement between
measurements and estimates based on GFs of oxalic acid from Mikhailov et al. (2009). The 1:1 $NH_4NO_3$/OA
mixed particles take up no water below 80% RH and exhibit a distinct deliquescence transition at about 86%
RH. It can be seen that the ZSR predictions based on our measurements fail to reproduce the hygroscopic
growth of $NH_4NO_3$/OA mixed particles with an equal mass ratio in the whole RH range. After full
deliquescence, the water contents associated with the 1:1 $NH_4NO_3$/OA mixed particles are comparable to the
ZSR predictions based on GFs of oxalic acid from Mikhailov et al. (2009), indicating that water uptake of
mixed particles is enhanced by dissolution of solid oxalic acid at high RH. It is obvious that the ammonium
nitrate in the initial mixed particles exists in a crystalline form rather than an amorphous one as in the pure
component particles. The possible reason is that the solid oxalic acid dihydrate seeds formed in the mixed
droplets during the drying process could trigger the heterogeneous nucleation of ammonium nitrate. The
previous study has reported that the presence of solid succinic acid seeds dramatically lowered the barrier to
promote the crystallisation of ammonium nitrate even when succinic acid content was only 25% by mass
(Lightstone et al., 2000). In contrast to ammonium nitrate/succinic acid mixtures, the deliquescence transition
of ammonium nitrate is not observed for the 3:1 $NH_4NO_3$/OA mixed particles, suggesting that the minor OA
fraction could not effectively initiate the crystallisation of ammonium nitrate. The hygroscopic behavior of
$NH_4NO_3$/OA mixed particles also reveals the crystallization of oxalic acid with the presence of hygroscopic
$NH_4NO_3$ at low RH. However, our previous study indicates that the coexisting hygroscopic organics such as

levoglucosan and malonic acid could hinder the crystallization of oxalic acid upon dehydration (Jing et al., 2016). It can be concluded that the interactions between $NH_4NO_3$ and oxalic acid have greater influence on phase state of $NH_4NO_3$ than oxalic acid. Compared to the hygroscopic growth of $NH_4NO_3$/OA mixed particles, oxalic acid has slight influence on the deliquescence transition of ammonium sulfate, as indicated by the slightly decreased deliquescence point of mixed particles relative to pure ammonium sulfate (Jing et al., 2016; Wang et al., 2017).

Figure 3 illustrates hygroscopic characteristics of $NH_4NO_3$ particles internally mixed with malonic acid (MA), succinic acid (SA) and phthalic acid (PA) at equal mass ratio, respectively. The measured hygroscopic growth of $NH_4NO_3$/MA mixed particles is in fair agreement with ZSR predictions. This suggests that the presence of malonic acid has no effect on the hygroscopic behavior of ammonium nitrate. However, the malonic acid could dramatically influence the deliquescence behavior of ammonium sulfate by promoting water uptake of mixed particles under low RH conditions (Prenni et al., 2003). For $NH_4NO_3$/SA mixed particles, no water uptake is observed until RH increases to 60%, consistent with the reported deliquescence point of $60.6 \pm 0.4\%$ for equal mass $NH_4NO_3$/SA mixed particles using the electrodynamic balance technology (Lightstone et al., 2000). It is obvious that succinic acid could influence phase behavior of ammonium nitrate. The crystallization of ammonium nitrate initiated by heterogeneous nucleation is favored by the presence of solid succinic acid seeds. The ZSR predictions underestimated the water uptake of $NH_4NO_3$/SA mixed particles above 80% RH. This phenomenon can be attributed to the partial dissolution of succinic acid, which thus contributed to water uptake by mixed aerosols at high RH. The early studies also found the enhanced hygroscopic growth of multicomponent aerosols containing succinic acid compared to ZSR estimates without taking limited solubility of succinic acid into account (Svenningsson et al., 2006; Wang et al., 2016b). For $NH_4NO_3$/OA and $NH_4NO_3$/SA mixed particles, the water uptake by the $NH_4NO_3$ component at high RH could trigger the partial or complete dissolution of oxalic acid and succinic acid. This behavior could be expected for a system at thermodynamic equilibrium, corresponding to a particle with an aqueous phase of the inorganic and organic composition, and solid organic acid (Clegg and Seinfeld, 2006). When the solid organic acids are in a metastable state, the complete dissolution could occur for aerosol particles composed of inorganic salt and organic acid above the deliquescence point of inorganic salt (Choi and Chan, 2002; Clegg and Seinfeld, 2006). Compared to ammonium nitrate, the deliquescence behavior and water uptake of ammonium sulfate are almost not affected by succinic acid (Prenni et al., 2003). The $NH_4NO_3$/PA mixed particles show continuous water uptake without obvious phase transitions over the whole RH range. At high RH, the ZSR rule could reproduce the GFs of mixed particles while significant deviation between measurements and predictions is observed at low and medium RH. The possible reason is that due to mass transfer limitation the residence time of 5 s

appears to be insufficient for 100 nm NH$_4$NO$_3$/PA aerosols to reach hygroscopic equilibrium in the low and
medium RH range.
**3.3 Water uptake behaviors of NaNO$_3$/organic acid mixtures**
Figure 4 shows the hygroscopic growth of 100 nm NaNO$_3$ particles internally mixed with oxalic acid, malonic
acid, succinic acid and phthalic acid at equal mass ratio, respectively. As can be seen in Fig. 4a, 1:1 NaNO$_3$/OA
mixed particles exhibit gradual water uptake without prompt deliquescence transition. The comparisons of
measurement results and ZSR predictions suggest that sodium nitrate still remains amorphous and thus
contributes to water uptake of mixed particles from low RH, which eventually results in the dissolution of
oxalic acid at high RH. In contrast to ammonium nitrate, the crystallization of sodium nitrate does not occur
even with the presence of oxalic acid at half mass fraction. For 1:1 NaNO$_3$/MA mixed particles, the continuous
hygroscopic growth of this mixed system agrees well with the ZSR predicted curve across all RH studied. The
prior study has indicated that the reaction between malonic acid and sodium nitrate within aerosols during the
dehydration process could lead to considerable nitrate depletion and formation of organic salts due to the
evaporation of HNO$_3$ (Wang and Laskin, 2014). However, it is clear that this reaction has negligible impacts on
the overall hygroscopic behavior of NaNO$_3$/MA aerosols, as indicated by the good agreement between
measured growth and ZSR predictions in the RH range studied. It can be explained by the fact that the
hygroscopic growth of sodium malonate is comparable to that of malonic acid at each given RH (Wu et al.,
2011). Disodium malonate has a GF of 1.78 at 90% RH (Wu et al., 2011). The field measurement has observed
the formation of sodium malonate in ambient aerosols (Laskin et al., 2012).
In the case of 1:1 NaNO$_3$/SA mixed particles (Fig. 4c), no distinct deliquescence transition was observed
upon hydration, which differed from the water uptake behavior of 1:1 NH$_4$NO$_3$/SA. Again, it shows sodium
nitrate is difficult to crystallize despite the presence of succinic acid that has a high deliquescence point. In
addition, the amount of water uptake by mixed particles is significantly larger than predictions from the ZSR
rule above 60% RH. The dissolution of succinic acid in aqueous NaNO$_3$ solution should be responsible for
enhanced water uptake of mixed particles. As shown in Fig. 4d, the equal mass NaNO$_3$/PA mixed aerosols also
present continuous water uptake from low RH. In contrast to 1:1 NaNO$_3$/SA mixed system, the dramatic
decrease in amount of water uptake by 1:1 NaNO$_3$/PA mixed system can be observed over the whole RH range
compared to ZSR estimates. This discrepancy appears to be caused by mass transfer limitation. It has been
found that after drying processing aerosol particles containing an inorganic salt and an organic acid with limited
solubility tend to remain phase separation with the organic phase distributed at the outer layer (Peckhaus et al.,
2012; Veghte et al., 2014). Zhou et al. (2014) observed the phase separation behavior of ammonium
sulfate/phthalic acid mixed particles due to the salting-out effect. It should be noted that phthalic acid has the

much lower solubility (0.41 g/100 mL) and oxygen-carbon ratio (O : C = 0.5) than other organic acids such as oxalic acid (9.5 g/100 mL, O : C = 2), malonic acid (76 g/100 mL, O : C = 1.33) and succinic acid (8.35 g/100 mL, O : C = 1). It has been found that a particle containing one dicarboxylic acid with limited solubility (< 1 g/100 mL) and ammonium sulfate would form phase-separated structures upon drying (Veghte et al., 2014). The prior studies observed that liquid–liquid phase separation never occurred for mixed systems containing organic species with the oxygen-to-carbon elemental ratio (O : C) larger than 0.8 and always occurred for O : C less than 0.5 (You et al., 2013). Considering the low solubility and oxygen-carbon ratio of phthalic acid, it can be expected the occurrence of phase-separated structures for phthalic acid-containing particles. As shown in Fig. S2, a cover layer can be observed at the border of the dry $NaNO_3$/phthalic acid mixed particle. The prior study for ammonium sulfate/succinic acid mixtures showed that mixed particles with core-shell structure had reduced hygroscopic growth relative to the well-mixed particles (Maskey et al., 2014). Similarly, the phthalic acid layer likely inhibits the water transfer from gas phase to $NaNO_3$ phase, thus resulting in the lower growth of particles.

### 3.4 Water uptake behaviors of $Ca(NO_3)_2$/organic acid mixtures

Due to the interference of precipitation, only hygroscopic growth of 100 nm $Ca(NO_3)_2$ particles internally mixed with malonic acid or phthalic acid at equal mass ratio is given in Fig. 5. The particle size for 1:1 $Ca(NO_3)_2$/MA aerosols upon hydration shows slight decrease between 10% and 40% RH and subsequently continuous growth without obvious phase transition in the whole RH range, which distinguishes from hygroscopic growth of pure $Ca(NO_3)_2$ and malonic acid. The similar hygroscopic behavior including the decrease in particle size at low RH was also observed for mixed $NH_4NO_3$/protein particles (Mikhailov et al., 2004) as well as amorphous oxalic acid and SOA particles derived from methylglyoxal-methylamine aqueous reactions upon hydration below 50% RH (Mikhailov et al., 2009; Hawkins et al., 2014). As shown in Fig. 6, the $Ca(NO_3)_2$/malonic acid (1:1) particle has a more compact sphere structure at 30% RH relative to the gel-like structure under dry conditions. This specific behavior of particle shrinkage upon hydration typically results from humidity-induced transformation of porous, gel-like structures into more compact sphere for (semi-)solid amorphous particles (Mikhailov et al., 2009). Thus, the hygroscopic characteristics of $Ca(NO_3)_2$/MA aerosols suggest the structure of mixed particles distinguishes from that of pure component particles.

The 1:1 $Ca(NO_3)_2$/PA aerosols take up water gradually with exposure of elevated RH. The overestimated water uptake by the ZSR rule could be observed below 80% RH. This substantial deviation between predictions and measurements can be attributed to mass transfer limitation in the aerosols (Chan and Chan, 2005). For the $Ca(NO_3)_2$/phthalic acid mixed particle, the clear organic cover layer at the border can be observed upon hydration, seen in Fig. S2. When the residence time of 5 s in the HTDMA measurements was expanded to 27 s, a slight increase (0.02-0.04) in GFs was observed for phthalic acid-containing mixed particles at 80% RH. The

outer organic layer may result in mass transfer limitations of water. Due to low molecular diffusivity, organic (semi-)solid amorphous structures at low RH could retard the equilibrium between evaporation and condensation of water from submicron particles on (multi-)second time scales (Booth et al., 2014; Price et al., 2014), thus kinetically inhibiting the hygroscopic growth of particles with insufficient equilibrium time (Mikhailov et al., 2009). When ambient humidity increases to high RH such as 80%, more water content in particles significantly reduces the impacts of mass transfer limitation, as indicated by the general agreement between predictions and measurements above 80% RH in Fig. 5. Although it has been found that nitrate depletions likely occur within $Ca(NO_3)_2$/organic acid aerosols, the reactions between $Ca(NO_3)_2$ and organic acids may have no obvious impacts on the hygroscopic growth in our experiment due to mass transfer limitation. Considering $Ca(NO_3)_2$/organic acid particles in the viscous semisolid state after dehydration (Wang and Laskin, 2014), the low molecular diffusivity in amorphous phases could limit release of $HNO_3$ from particle phase on (multi-)second time scales. In addition, our previous study also showed that due to the presence of coexisting hygroscopic species the transformation of solid state into viscous semisolid state for NaCl/oxalic acid mixed particles considerably inhibited chloride depletions or HCl evaporation during the dehydration process (Peng et al., 2016).

## 4 Conclusions and atmospheric implications

Our results reveal that the nitrate salt/organic acid mixed aerosols exhibit varying phase behavior and hygroscopic growth depending upon the type of components present in the particles. Whereas pure nitrate salt particles show continuous water uptake with increasing RH, the deliquescence transition is still observed for ammonium nitrate particles internally mixed with organic acids such as oxalic acid and succinic acid with a high deliquescence point. In contrast to ammonium nitrate, hygroscopic nature of submicron sodium nitrate aerosols is characterized by continuous growth even with the presence of oxalic acid or succinic acid, indicating that sodium nitrate particles tend to exist in a liquid-like state under near dry conditions. Although pure oxalic and succinic acids exhibit no hygroscopic growth in the RH range studied (0-90%), the nitrate salt particles mixed with oxalic or succinic acid have enhanced hygroscopic growth at high RH due to water uptake by dissolved fractions of oxalic or succinic acid. In the case of calcium nitrate particles containing malonic acid or phthalic acid, the water uptake of mixed aerosols is significantly inhibited in the low and moderate RH range likely due to mass transfer limitation in amorphous solid with high viscosity. Our findings indicate that the coexisting organic acids modify the phase and morphology of nitrate salt particles in the low and medium RH range, which thus likely result in obvious enhancement or suppression of water uptake with further increasing RH.

Due to the enhanced fractions of nitrate in particulate matter and controlled sulfate by policies (Hodas et al.,

2014), the nitrate salt may play an important role in the hygroscopicity and thus the water content of aerosol particles in urban areas such as Eastern United States and Eastern China where nitrate concentrations are high (Zhang et al., 2015). The phase state and water uptake of atmospheric particles have crucial impacts on determining the role of aerosols in earth's climate and air quality. The well understanding of the phase state of atmospheric particles could reduce the uncertainty in radiative forcing estimates. The new findings for phase behavior of nitrate salts aid in further understanding the atmospheric lifetime, optical properties, and cloud formation of nitrate salts. In addition, these experimental results can serve as a basis for the evaluation and improvement of thermodynamic models for prediction of aerosol's physicochemical properties. The ongoing effort to understand the mechanisms of interactions between water and aerosol particles with varying compositions is of importance in incorporating these processes into global climate models.

**Data availability.** All data are available upon request from Bo Jing (jingbo109@163.com) or Maofa Ge (gemaofa@iccas.ac.cn).

**Competing interests.** The authors declare that they have no conflict of interest.

**Author contribution.** BJ and MG designed the experiments. BJ and ZW carried them out. BJ performed the data analysis and prepared the manuscript with contributions from all co-authors.

**Acknowledgment**

This project was supported by the National Natural Science Foundation of China (Contract No. 41227805, 91544227, 21477134) and the National Key Research and Development Program of China (2016YFC0202202). The authors would like to thank Chao Peng, Ying Zhang and Kaihui Xia for help in additional measurements during the revision period.

**The Supplement related to this article is available online at supplement.**

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

1    **Table 1.** Chemical properties of substances investigated in this study.

| Chemical compounds | Molecular formula | Molar weight [g mol$^{-1}$] | Density [g cm$^{-3}$] | Solubility [g 100 cm$^{-3}$ H$_2$O][a] | Manufacturer/Purity |
|---|---|---|---|---|---|
| Ammonium nitrate | $NH_4NO_3$ | 80.0 | 1.72 | 213 | Dtftchem, 99% |
| Sodium nitrate | $NaNO_3$ | 85.0 | 2.26 | 91.2 | Alfa Aesar, ≥99.0% |
| Calcium nitrate | $Ca(NO_3)_2$ | 164.1 | 2.5 | 144 | |
| Calcium nitrate tetrahydrate | $Ca(NO_3)_2 \cdot 4H_2O$ | 236.1 | 1.82 | 144 | Alfa Aesar, 99.0% |
| Oxalic acid | $C_2H_2O_4$ | 90.0 | 1.90 | 9.5 | Aldrich, 99.999% |
| Oxalic acid (dihydrate) | $C_2H_2O_4 \cdot 2H_2O$ | 126.1 | 1.65 | | |
| Malonic acid | $C_3H_4O_4$ | 104.1 | 1.62 | 76 | Sigma-Aldrich, 99% |
| Succinic acid | $C_4H_6O_4$ | 118.1 | 1.57 | 8.35 | Sigma-Aldrich, ≥99.5% |
| Phthalic acid | $C_8H_6O_4$ | 166.1 | 1.59 | 0.41[b] | Sigma-Aldrich, ≥99.5% |

2    [a] From *CRC Handbook of Chemistry and Physics* at 298 K; [b] From Hartz et al. (2006)

**Table 2.** Hygroscopic growth factors of nitrate salt containing particles in this study. The predictions for pure
component and mixtures are from the AIOMFAC and ZSR, respectively. For oxalic acid containing mixtures,
the predicted values based on measurements from Mikhailov et al. (2009) are shown in bracket.

| | GF (80% RH) | | | GF (90% RH) | | |
|---|---|---|---|---|---|---|
| | Measured | Predicted | Literature | Measured | Predicted | Literature |
| **Pure component** | | | | | | |
| $NH_4NO_3$ | 1.42 | 1.41 | 1.40[a] | 1.70 | 1.72 | 1.75[a] |
| $NaNO_3$ | 1.62 | 1.63 | 1.60[b] | 1.97 | 1.95 | 1.91[b] |
| $Ca(NO_3)_2$ | 1.56 | 1.52 | 1.51[c] | 1.89 | 1.82 | NA |
| **Equal mass mixtures** | | | | | | |
| $NH_4NO_3$ / oxalic acid | 1.07 | 1.23 (1.33) | NA | 1.64 | 1.44 (1.65) | NA |
| $NH_4NO_3$ / malonic acid | 1.41 | 1.38 | NA | 1.91 | 1.81 | NA |
| $NH_4NO_3$ / succinic acid | 1.26 | 1.24 | NA | 1.74 | 1.52 | NA |
| $NH_4NO_3$ / phthalic acid | 1.27 | 1.28 | NA | 1.69 | 1.62 | NA |
| $NaNO_3$ / oxalic acid | 1.37 | 1.27 (1.38) | NA | 1.79 | 1.53 (1.75) | NA |
| $NaNO_3$ / malonic acid | 1.44 | 1.47 | NA | 2.02 | 1.93 | NA |
| $NaNO_3$ / succinic acid | 1.52 | 1.32 | NA | 2.09 | 1.68 | NA |
| $NaNO_3$ / phthalic acid | 1.22 | 1.37 | NA | 1.61 | 1.68 | NA |
| $Ca(NO_3)_2$ / malonic acid | 1.32 | 1.44 | NA | 1.87 | 1.77 | NA |
| $Ca(NO_3)_2$ / phthalic acid | 1.28 | 1.37 | NA | 1.93 | 1.65 | NA |

[a] Hu et al. (2011).
[b] Gysel et al. (2002).
[c] Gibson et al. (2006).
NA: not available.

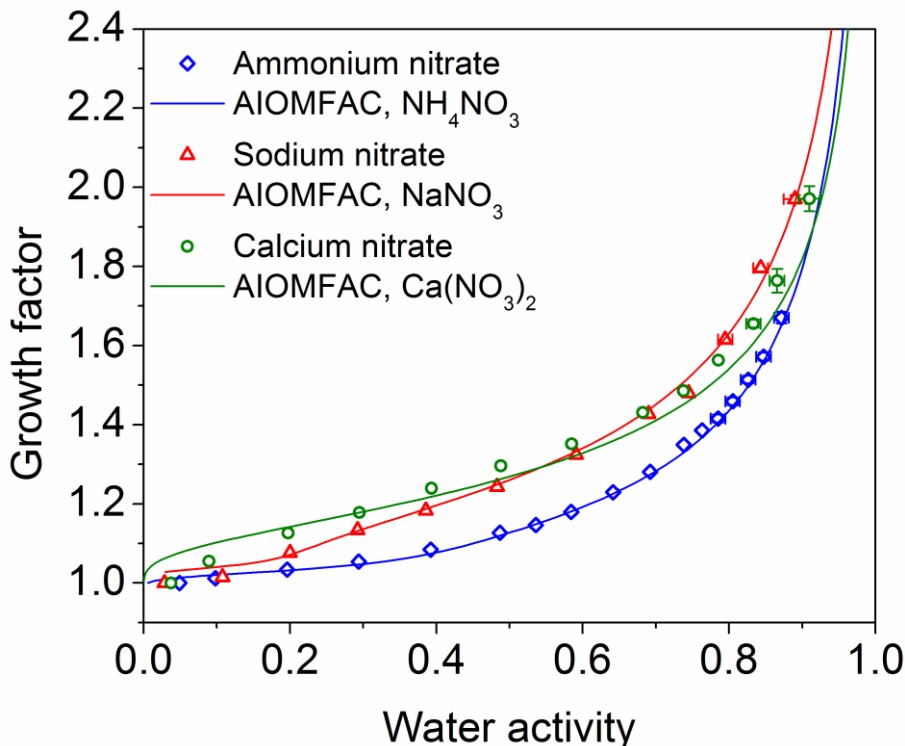

**Figure 1.** Hygroscopic growth factors of 100 nm $NH_4NO_3$, $NaNO_3$, $Ca(NO_3)_2$ particles as a function of water
activity. The corresponding AIOMFAC-predicted curve associated with measurements is also presented. The
error bars representing the standard deviation of multiple measurements are generally not obvious compared to
the size of the symbols.

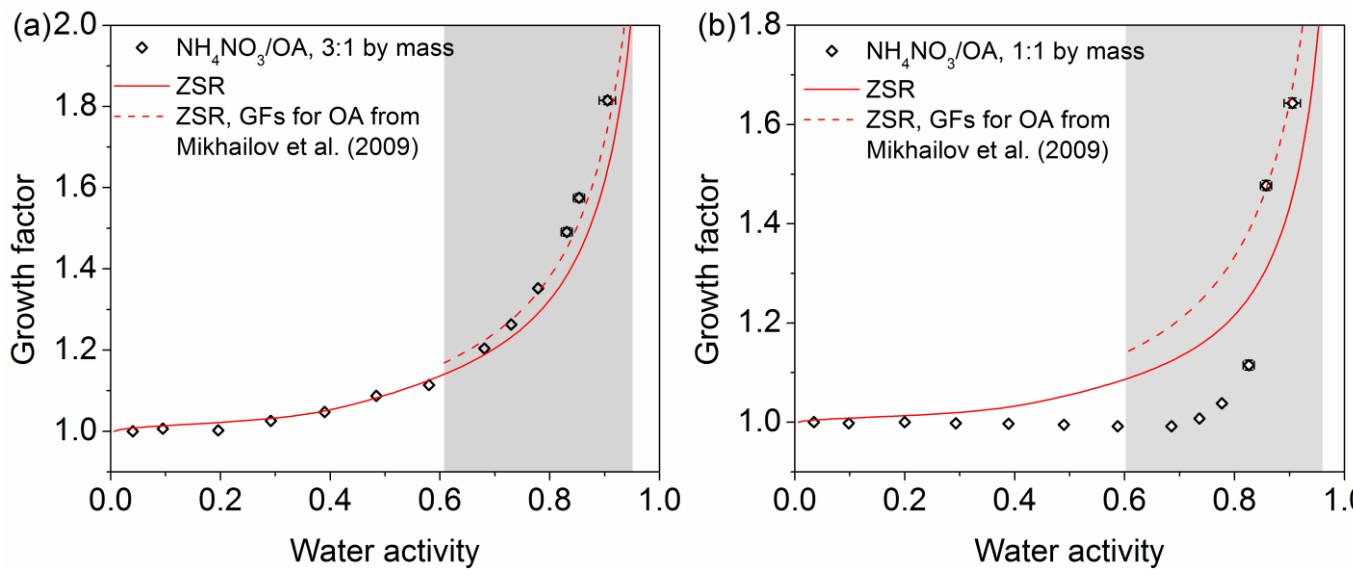

**Figure 2.** Hygroscopic growth factors of 100 nm NH$_4$NO$_3$/oxalic acid (OA) mixed particles with mass ratios of
(a) 3:1 and (b) 1:1 as a function of water activity. The ZSR-predicted curves based on GF = 1 for oxalic acid or
GFs from Mikhailov et al. (2009) are indicated by red solid and dash line, respectively. The error bars
representing the standard deviation of multiple measurements are generally not obvious compared to the size of
the symbols. The shading area shows potential water uptake by oxalic acid component under high RH
conditions.

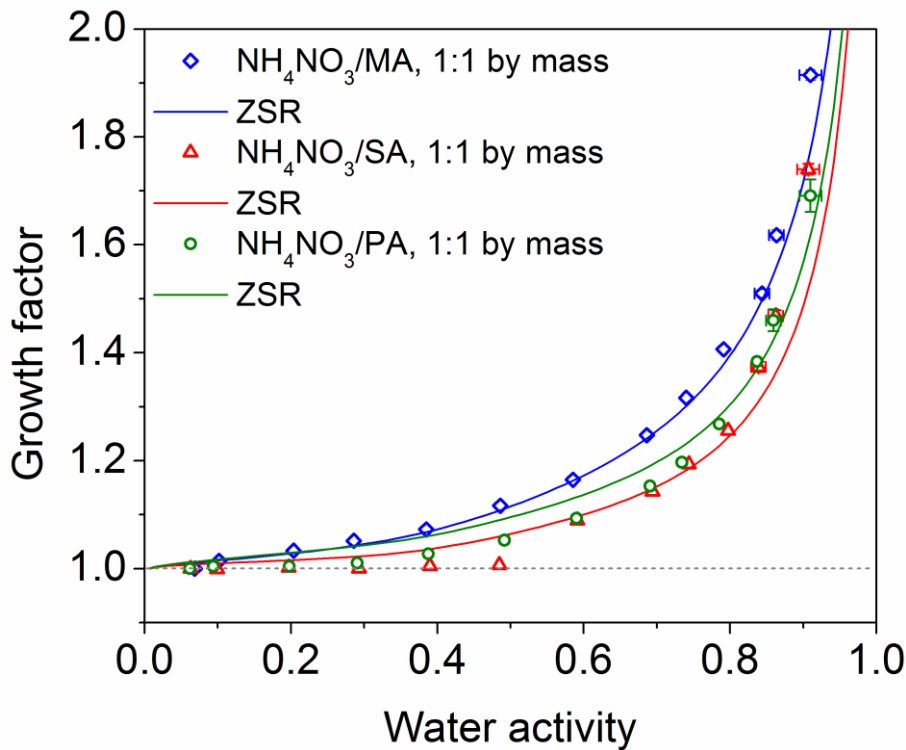

**Figure 3.** Hygroscopic growth factors of 100 nm internally mixed particles composed of $NH_4NO_3$/malonic acid (MA), $NH_4NO_3$/succinic acid (SA) and $NH_4NO_3$/phthalic acid (PA) with equal mass ratios as a function of water activity. The blue, red and green solid line indicates the hygroscopic growth predicted from the ZSR method. The error bars representing the standard deviation of multiple measurements are generally not obvious compared to the size of the symbols.

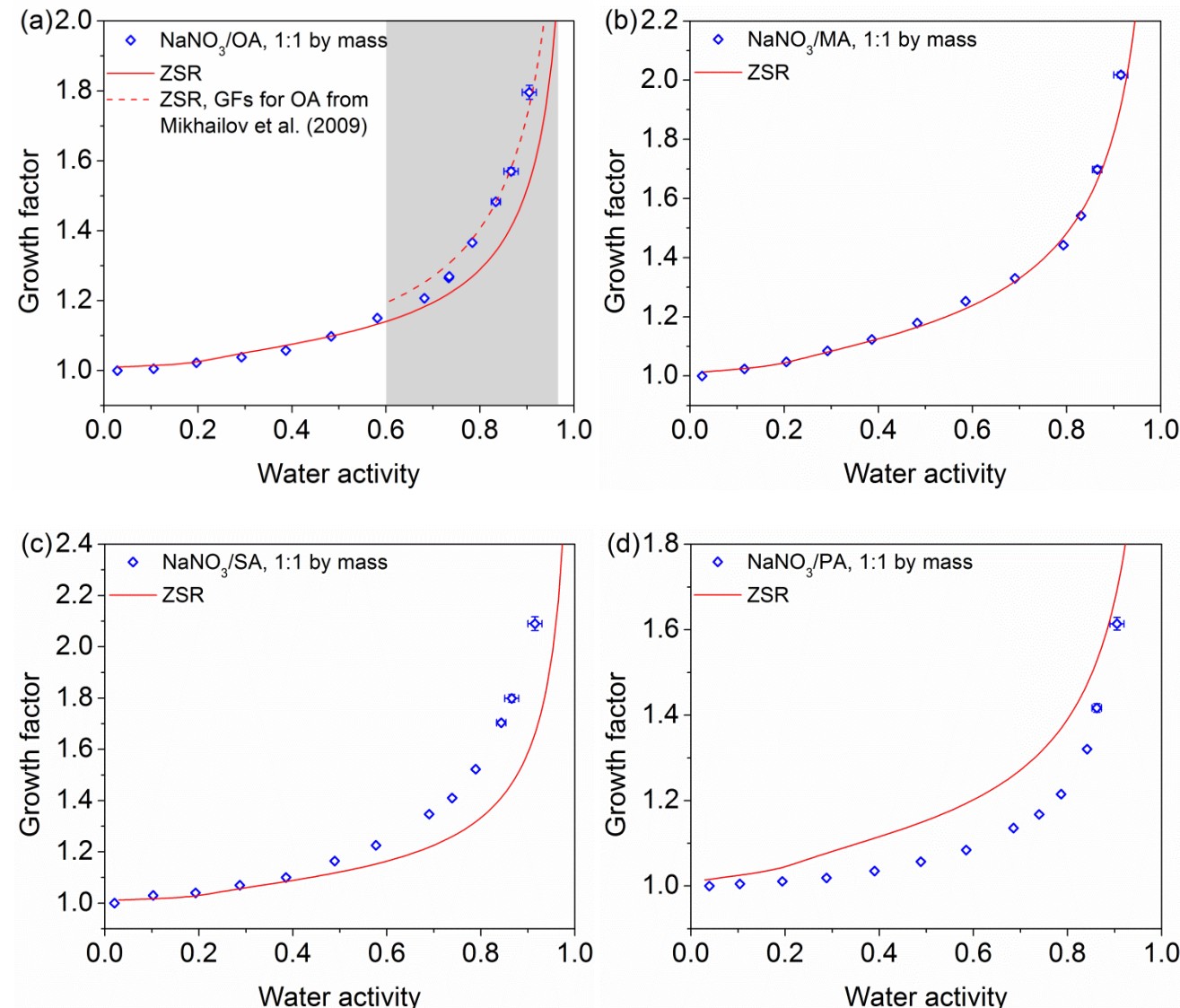

**Figure 4.** Hygroscopic growth factors of 100 nm internally mixed particles composed of NaNO$_3$/oxalic acid (OA) (a), NaNO$_3$/malonic acid (MA) (b), NaNO$_3$/succinic acid (SA) (c) and NaNO$_3$/phthalic acid (PA) (d) with equal mass ratios as a function of water activity. The red line indicates the hygroscopic growth predicted from the ZSR method. For the NaNO$_3$/oxalic acid (OA) mixture, the ZSR-predicted curve based on the GFs for OA from Mikhailov et al. (2009) is presented as the dash line. The shading area shows potential water uptake by oxalic acid component. The error bars representing the standard deviation of multiple measurements are not obvious compared to the size of the symbols.

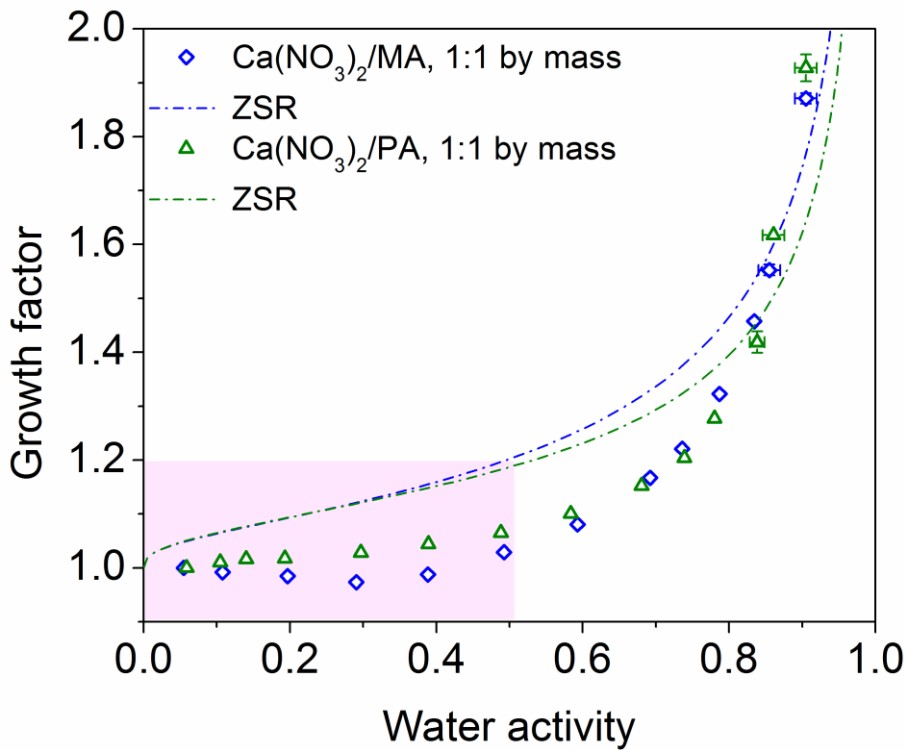

**Figure 5.** Hygroscopic growth factors of 100 nm internally mixed particles composed of $Ca(NO_3)_2$ with
malonic acid (MA) or phthalic acid (PA) at equal mass ratios as a function of water activity. The blue and green
dash dot line indicates the hygroscopic growth predicted from the ZSR method. The error bars representing the
standard deviation of multiple measurements are not obvious compared to the size of the symbols. The shading
area shows no significant water uptake by the mixed systems under low RH conditions.

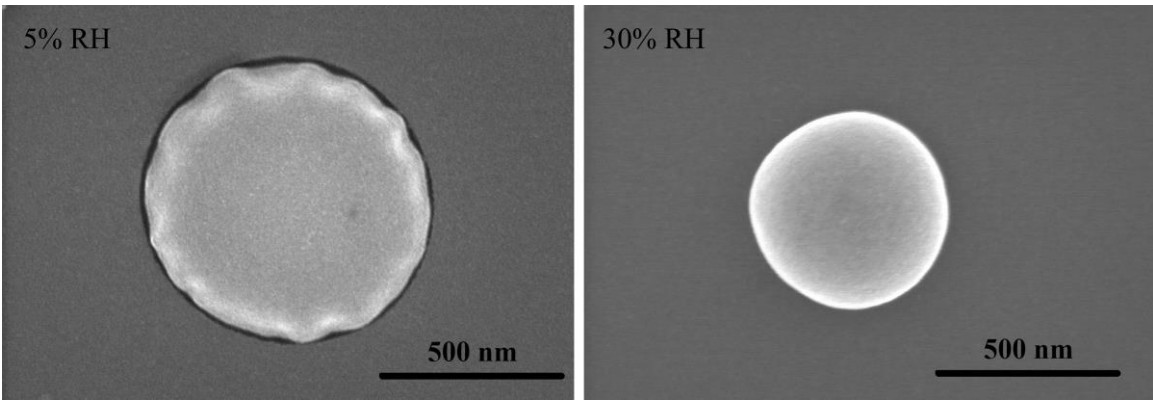

**Figure 6.** SEM micrographs of deposited particles composed of calcium nitrate and malonic acid before conditioning (left) and after conditioning at 30% RH (right).