# Peer review of "Hygroscopic behavior of atmospheric aerosols containing"

_Atmospheric Chemistry and Physics, 2017_

## Referee Comment (RC1) · Anonymous Referee #1 · 21 Dec 2017

The present manuscript focussed on the hygroscopic behaviors of some of the nitrate aerosols and their mixtures with water-soluble organic acids at different fractions using a hygroscopic tandem DMA. The authors explained the hygroscopic behaviors of all investigated salts and mixtures by comparing with ZSR model and some previous studies. They explained the difference between the measured and predicted growth factors in terms of initial phase state and mass transfer limitation. The authors have published several papers on hygroscopic behaviors of various compounds and their mixing with water-soluble organic acids, hence; methodology and data analysis are trustworthy and sound. However, there are many speculations and biased statements in the manuscript. The authors have drawn conclusions without providing evidence. It is hard to believe for the same reasons (dissolution and mass transfer limitation) to the

variation in growth factors of all investigated substances of this study. Except for these two reasons, could not find any novelty in this study. Unfortunately, the manuscript in its present form does not meet the standards of the journal, thus, unable to recommend for the publication in the community of ACP.

Major comments The authors explained hygroscopic behaviors of all substances based on suspected phase state and mass transfer limitations without providing any evidence. They just assumed dissolution of organic acids if the growth factor of nitrate salts-organic mixtures higher than the predicted ZSR value. In contrast, lower growth factors of nitrate –organic mixtures are interpreted with mass transfer limitations. These are the only possible explanations for all investigated mixtures in this study. I suggest the authors should provide more reliable information in order to assert their conclusions that how the coating of organics changes the phase state of nitrate aerosols? Are there any additional measurements to show the phase state of each system? TEM analysis?

Insufficient residence time and mass transfer limitation– why only for NH4NO3/PA and other calcium nitrate mixed aerosols and why not for others? Authors should discuss more about this issue. The authors should show more evidence for their conclusion drawn as insufficient residence time for above mentioned mixed particles.

Although the authors had stated in the abstract that they investigated mixtures with varying organic fractions, but showed only one fraction (1:1) for all the mixed systems in the manuscript. How dissolution happened for NH4NO3/OA and NH4NO3/SA mixed particles. The authors should discuss more about this function because this is the only reason for observed higher growth factors of those particles compared to predicted. What is the exact cause of dissolution? Why OA did not show dissolution in this study. How much it is correct to use the growth factor of OA from the study of Mikhailov et al. (2009) in ZSR model of present study.

The authors should report all the measured and predicted growth factors in a Table

in addition to literature values so that the readers can easily follow the text in the manuscript.

Specific comments P6L9: The authors should replace nitrates with nitrate salts here as well as in the whole manuscript. P6L21: The authors should report the measured and predicted hygroscopic growth factor of $Ca(NO_3)_2$ particles here. P6L22: Here, the assumption of phase state (for example, amorphous) just because of existing water-uptake at lower RH is sometimes revolting although it is the main conclusion of their study. I think the author should show some more evidence for phase state rather based on previous studies and water uptake at lower RH. Contamination also plays a role on water uptake at lower RH. P7L10-24: Should move these lines (about discussion of ZSR and AIOMFAC models) to section 2. P7L5-6: Malonic acid and phthalic acid exhibited continuous-water uptake in whole RH rage as the authors stated. This means the initial phase state of these acids is amorphous? P9L18-19: Report the growth factor value of sodium malonate here and also cite a reference about the formation of sodium acetate in ambient aerosols. P10L15-19: This statement is biased and no reasonable evidence for observed particle shrinkage. Do you have any electronic images like TEM etc.; it is hard to believe that the existence of gel-like structures in this study without showing any electronic images.

The authors must and should provide more shreds of evidence for their conclusions.

---

## Referee Comment (RC2) · Anonymous Referee #2 · 24 Dec 2017

Jing et al. characterized the hygroscopicity of nitrate aerosols and their mixtures with several model organic acids using an HTMDA. The authors show that the presence of the organic acids can alter the phase behaviour and water uptake of nitrate aerosols, resulting in unexpected growth relative to the modelled growth. This study provides a set of valuable data for the hygroscopic behaviour of nitrate-containing aerosols generated in the lab, which has important implications for the understanding of physicochemical behaviours of atmospherically nitrate-containing particles. The data in this study appear to be of good quality. This manuscript is well organized and clearly presented. It is recommended for publication in ACP after the following minor comments are addressed. 1. P3 L14: Could the authors quantify the nitrate content in the atmospheric particulate matter? 2. P6 L5: Please add a quantitative comparison to literature val-

ues. 3. P6 L23: The authors can convert the GFs of pure nitrate particles into kappa values based on the method proposed by Petters and Kreidenweis (ACP, 2007). Such a comparison between this study and literature would likely make the results more useful to a broader audience. 4. P6 L27, Figure 1: It seems that AIOMFAC performs better for ammonium and sodium nitrates than calcium nitrates. Please elaborate on the possible reasons. 5. Can the authors explain whether AIOMFAC predictions are from fitted data or from an ab initio prediction? If AIOMFAC is from a fit to data, please cite that dataset. 6. P7 L6: Please quantify how your results are consistent with the observations in the literature. 7. P7 L8, section 3.2: Comparison with the literature results of ammonium sulphate is encouraged when appropriate. 8. P9 L30-31: Please quantify the limited solubility. The occurrence of liquid-liquid phase separation is also influenced by the oxygen-carbon ratio of organic components. If possible, please expand the discussion regarding liquid-liquid phase separation. Thus, it can be helpful to the readers who are not familiar with this issue. 9. P11 L25: The findings reported in this study are likely significant for areas where nitrate concentrations are high. The authors may consider adding some specific areas where your suggested phenomena would be more pronounced.

---

## Author Comment (AC1) · 13 Feb 2018

**Author's Response**

**Response to Referee #1:**

We are grateful for the reviewer's comments. Those comments are valuable and helpful for improving our paper. Our response to the comments and corresponding changes in the manuscript are included below. We repeat the specific points raised by the reviewer in bold font, followed by our response in italic font. The pages numbers and lines mentioned below are consistent with those in the Atmospheric Chemistry and Physics Discussions (ACPD) paper.

**The present manuscript focussed on the hygroscopic behaviors of some of the nitrate aerosols and their mixtures with water-soluble organic acids at different fractions using a hygroscopic tandem DMA. The authors explained the hygroscopic behaviors of all investigated salts and mixtures by comparing with ZSR model and some previous studies. They explained the difference between the measured and predicted growth factors in terms of initial phase state and mass transfer limitation. The authors have published several papers on hygroscopic behaviors of various compounds and their mixing with water-soluble organic acids, hence; methodology and data analysis are trustworthy and sound. However, there are many speculations and biased statements in the manuscript. The authors have drawn conclusions without providing evidence. It is hard to believe for the same reasons (dissolution and mass transfer limitation) to the variation in growth factors of all investigated substances of this study. Except for these two reasons, could not find any novelty in this study. Unfortunately, the manuscript in its present form does not meet the standards of the journal, thus, unable to recommend for the publication in the community of ACP.**

*Reply: We thank the reviewer for the comments. According to the reviewer's suggestion, we provide experimental evidences and supplement some discussions to further support our conclusions. The detailed response including supplement and modification can be seen below.*

**Major comments**

**The authors explained hygroscopic behaviors of all substances based on suspected phase state and mass transfer limitations without providing any evidence. They just assumed dissolution of organic acids if the growth factor of nitrate salts-organic mixtures higher than the predicted ZSR value. In contrast, lower growth factors of nitrate–organic mixtures are**

interpreted with mass transfer limitations. These are the only possible explanations for all investigated mixtures in this study. I suggest the authors should provide more reliable information in order to assert their conclusions that how the coating of organics changes the phase state of nitrate aerosols? Are there any additional measurements to show the phase state of each system? TEM analysis?

*Reply: We thank the reviewer for the suggestion. We understand that some readers may wonder how the phase state of aerosols is determined without any visual measurements. In fact, the measured hygroscopicity curve of aerosols itself could effectively characterize the phase state and phase change of particles. This method has been widely applied in the hygroscopicity studies by many prestigious groups in atmospheric (chemistry and physics) community (Hemming and Seinfeld, 2001; Cruz and Pandis, 2000; Choi and Chan, 2002; Gysel et al., 2002; Prenni et al., 2003; Mochida and Kawamura, 2004; Mikhailov et al., 2009; Hodas et al., 2015; Estillore et al., 2016). Pöschl et al. have concluded that amorphous substances tend to absorb water vapor and undergo gradual hygroscopic growth from low relative humidity (Mikhailov et al., 2009). Depending on viscosity and microstructure, the amorphous phases can be classified as glasses, rubbers, gels or viscous liquids (Mikhailov et al., 2009). The hygroscopicity studies mentioned above also proved that smooth or continuous water-uptake behavior without prompt deliquescence transitions during humidification is characteristic of that of amorphous, noncrystalline aerosol particles.*

*According to the reviewer's suggestion, we provide some additional experimental evidences to further support our conclusions, which can be seen in the following specific reply.*

**Insufficient residence time and mass transfer limitation– why only for NH4NO3/PA and other calcium nitrate mixed aerosols and why not for others? Authors should discuss more about this issue. The authors should show more evidence for their conclusion drawn as insufficient residence time for above mentioned mixed particles.**

*Reply: We appreciate the reviewer's suggestion. The reason why phthalic acid-containing mixed particles show mass transfer limitation is likely due to the distinct properties of phthalic acid ($C_8H_6O_4$) relative to other organic acids studied in our work. It should be noted that phthalic acid has the much lower solubility (0.41 g/100 mL) and oxygen-carbon ratio (O : C = 0.5) than other organic acids such as oxalic acid (9.5 g/100 mL, O : C = 2), malonic acid (76 g/100 mL, O : C =*

*1.33) and succinic acid (8.35 g/100 mL, O : C = 1). It has been found that a particle containing one dicarboxylic acid with limited solubility (< 1 g/100 mL) and ammonium sulfate would form phase-separated structures upon drying (Veghte et al., 2014). The prior studies observed that liquid–liquid phase separation never occurred for mixed systems containing organic species with the oxygen-to-carbon elemental ratio (O : C) larger than 0.8 and always occurred for O : C less than 0.5 (You et al., 2013). Considering the low solubility and oxygen-carbon ratio of phthalic acid, it can be expected the occurrence of phase-separated structures for phthalic acid-containing particles. As shown in Figure S2, a cover layer can be observed at the border of the dry NaNO₃/phthalic acid mixed particle. For the Ca(NO₃)₂/phthalic acid mixed particle, the clear organic cover layer at the border can be observed upon hydration. When the residence time of 5 s in the HTDMA measurements was expanded to 27 s, a slight increase (0.02-0.04) in GFs was observed for phthalic acid-containing mixed particles at 80% RH. The outer organic layer may result in mass transfer limitations of water.*

[Figure]

***Figure S2.** (a) SEM micrographs of the NaNO₃/phthalic acid mixed particle conditioned under dry conditions. Optical micrographs of the Ca(NO₃)₂/phthalic acid mixed particle at 6% RH (b) and 77%*

*RH (c) upon hydration.*

***Related changes in the revised manuscript:***

*Figure S2 is added into the supplement.*

***Page 10 line 1-2:*** *The sentence "Considering phthalic acid with limited solubility, the phase separation behavior may also occur for NaNO₃/PA mixed particles upon dehydration."* ***is changed into*** *"It should be noted that phthalic acid has the much lower solubility (0.41 g/100 mL) and oxygen-carbon ratio (O : C = 0.5) than other organic acids such as oxalic acid (9.5 g/100 mL, O : C = 2), malonic acid (76 g/100 mL, O : C = 1.33) and succinic acid (8.35 g/100 mL, O : C = 1). It has been found that a particle containing one dicarboxylic acid with limited solubility (< 1 g/100 mL) and ammonium sulfate would form phase-separated structures upon drying (Veghte et al., 2014). The prior studies observed that liquid–liquid phase separation never occurred for mixed systems containing organic species with the oxygen-to-carbon elemental ratio (O : C) larger than 0.8 and always occurred for O : C less than 0.5 (You et al., 2013). Considering the low solubility and oxygen-carbon ratio of phthalic acid, it can be expected the occurrence of phase-separated structures for phthalic acid-containing particles. As shown in Fig. S2, a cover layer can be observed at the border of the dry NaNO₃/phthalic acid mixed particle.".*

***Page 10 line 22: We add*** *"For the Ca(NO₃)₂/phthalic acid mixed particle, the clear organic cover layer at the border can be observed upon hydration, seen in Fig. S2. When the residence time of 5 s in the HTDMA measurements was expanded to 27 s, a slight increase (0.02-0.04) in GFs was observed for phthalic acid-containing mixed particles at 80% RH. The outer organic layer may result in mass transfer limitations of water.".*

**Although the authors had stated in the abstract that they investigated mixtures with varying organic fractions, but showed only one fraction (1:1) for all the mixed systems in the manuscript. How dissolution happened for NH4NO3/OA and NH4NO3/SA mixed particles. The authors should discuss more about this function because this is the only reason for observed higher growth factors of those particles compared to predicted. What is the exact cause of dissolution? Why OA did not show dissolution in this study. How much it is correct to use the growth factor of OA from the study of Mikhailov et al. (2009) in ZSR model of present study.**

***Reply:*** *We showed only one fraction (1:1) for all the mixed systems except for $NH_4NO_3$ / oxalic acid (OA) mixture (1:1, 3:1). To avoid confusion, P2L4: we remove "at varying mass ratios".*

*For $NH_4NO_3$/OA and $NH_4NO_3$/SA mixed particles, the water uptake by the $NH_4NO_3$ component at high RH could trigger the partial or complete dissolution of oxalic acid and succinic acid. This behavior could be expected for a system at thermodynamic equilibrium, corresponding to a particle with an aqueous phase of the inorganic and organic composition, and solid organic acid (Clegg and Seinfeld, 2006). When the solid organic acids are in a metastable state, the complete dissolution could occur for aerosol particles composed of inorganic salt and organic acid above the deliquescence point of inorganic salt (Choi and Chan, 2002; Clegg and Seinfeld, 2006).*

*As the pure oxalic acid particle exists as crystalline dihydrate, it shows no hygroscopic growth in our study. Without the influence of coexisting species, crystalline oxalic acid would not show dissolution below its deliquescence point (>97% RH). It should be noted that we used the growth factor of OA from the dehydration curve measured by Mikhailov et al. (2009). The dehydration curve at high RH corresponds to the supersaturated droplet of oxalic acid. The growth factor of OA from the dehydration curve used in our ZSR estimates could describe the dissolution of oxalic acid. In fact, our previous study has indicated that using the growth factor of OA from Mikhailov et al. (2009) in ZSR estimates could produce consistent results with that from thermodynamic methods such as AIOMFAC (Jing et al., 2017).*

***Related changes in the revised manuscript:***

***Page 2 line 4:*** *We remove "at varying mass ratios".*

***Page 8 line 31: We add*** *"For $NH_4NO_3$/OA and $NH_4NO_3$/SA mixed particles, the water uptake by the $NH_4NO_3$ component at high RH could trigger the partial or complete dissolution of oxalic acid and succinic acid. This behavior could be expected for a system at thermodynamic equilibrium, corresponding to a particle with an aqueous phase of the inorganic and organic composition, and solid organic acid (Clegg and Seinfeld, 2006). When the solid organic acids are in a metastable state, the complete dissolution could occur for aerosol particles composed of inorganic salt and organic acid above the deliquescence point of inorganic salt (Choi and Chan, 2002; Clegg and Seinfeld, 2006).".*

***Page 7 line 2:*** *"stable OA dihydrate"* ***is changed into*** *"crystalline OA dihydrate".*

***Page 7 line 3:*** *"The GFs of aqueous OA reported by Mikhailov et al. (2009)"* ***is changed into*** *"The*

*GFs of aqueous OA from the dehydration curve measured by Mikhailov et al. (2009)".*

**The authors should report all the measured and predicted growth factors in a Table in addition to literature values so that the readers can easily follow the text in the manuscript.**

*Reply: Thanks. According to the reviewer's suggestion, we add a Table containing measured and predicted growth factors as well as literature values.*

*Page 6 line 9: We add "Table 2 summarizes hygroscopic growth factors of nitrate containing particles in this study.".*

**Table 2.** Hygroscopic growth factors of nitrate containing particles in this study. The predictions for pure component and mixtures are from the AIOMFAC and ZSR, respectively. For oxalic acid containing mixtures, the predicted values based on measurements from Mikhailov et al. (2009) are shown in bracket.

| | GF (80% RH) | | | GF (90% RH) | | |
|---|---|---|---|---|---|---|
| | Measured | Predicted | Literature | Measured | Predicted | Literature |
| **Pure component** | | | | | | |
| $NH_4NO_3$ | 1.42 | 1.41 | 1.40[a] | 1.70 | 1.72 | 1.75[a] |
| $NaNO_3$ | 1.62 | 1.63 | 1.60[b] | 1.97 | 1.95 | 1.91[b] |
| $Ca(NO_3)_2$ | 1.56 | 1.52 | 1.51[c] | 1.89 | 1.82 | NA |
| **Equal mass mixtures** | | | | | | |
| $NH_4NO_3$ / oxalic acid | 1.07 | 1.23 (1.33) | NA | 1.64 | 1.44 (1.65) | NA |
| $NH_4NO_3$ / malonic acid | 1.41 | 1.38 | NA | 1.91 | 1.81 | NA |
| $NH_4NO_3$ / succinic acid | 1.26 | 1.24 | NA | 1.74 | 1.52 | NA |
| $NH_4NO_3$ / phthalic acid | 1.27 | 1.28 | NA | 1.69 | 1.62 | NA |
| $NaNO_3$ / oxalic acid | 1.37 | 1.27 (1.38) | NA | 1.79 | 1.53 (1.75) | NA |
| $NaNO_3$ / malonic acid | 1.44 | 1.47 | NA | 2.02 | 1.93 | NA |
| $NaNO_3$ / succinic acid | 1.52 | 1.32 | NA | 2.09 | 1.68 | NA |
| $NaNO_3$ / phthalic acid | 1.22 | 1.37 | NA | 1.61 | 1.68 | NA |
| $Ca(NO_3)_2$ / malonic acid | 1.32 | 1.44 | NA | 1.87 | 1.77 | NA |
| $Ca(NO_3)_2$ / phthalic acid | 1.28 | 1.37 | NA | 1.93 | 1.65 | NA |

[a] Hu et al. (2011).

[b] Gysel et al. (2002).

[c] Gibson et al. (2006).

NA: not available.

**Specific comments**

**P6L9: The authors should replace nitrates with nitrate salts here as well as in the whole manuscript.**

*Reply: We have replaced nitrates with nitrate salts in the whole revised manuscript.*

**P6L21: The authors should report the measured and predicted hygroscopic growth factor of Ca(NO3)2 particles here.**

*Reply: P6L23, **we add** "The GF of 100 nm $Ca(NO_3)_2$ particles is determined to be 1.56 and 1.89 at 80% and 90% RH, respectively, consistent with the corresponding predicted value of 1.52 and 1.82." in the new version.*

**P6L22: Here, the assumption of phase state (for example, amorphous) just because of existing water uptake at lower RH is sometimes revolting although it is the main conclusion of their study. I think the author should show some more evidence for phase state rather based on previous studies and water uptake at lower RH. Contamination also plays a role on water uptake at lower RH.**

*Reply: Thanks for the comments. The assumption of phase state (for example, amorphous) because of existing water uptake at lower RH is reasonable based on literature knowledge and our additional measurements. It is unbelievable that contamination also plays a role on water uptake at lower RH in our studies. If the water uptake of lab-generated aerosols at lower RH can also be explained by the influence of contamination, then the analysis and conclusions in most of the previous hygroscopicity studies by many scientific groups would be overturned. Since high pure reagents ($\geq$ 99.0%), ultrapure water ($\geq$ 18.2 M$\Omega$ cm) and high pure carrier gas ($N_2$, $\geq$ 99.99%) are used for aerosol preparation, it is reasonable to eliminate the influence of contamination on water uptake at lower RH. In fact, if the "contamination" exists and influence the water uptake of the aerosol particles in our study, why it did not induce the water uptake of succinic acid, $NH_4NO_3$/oxalic acid (1:1, by mass) and $NH_4NO_3$/succinic acid (1:1, by mass) particles at low RH,*

*as shown in Figures 2(b), 3 and S1(c).*

*As shown in Fig. S3, the nitrate salt particles stay in a liquid-like state at low RH less than 10%. The Raman spectrum of ammonium nitrate shows no obvious absorption between 3400 cm$^{-1}$ and 3600 cm$^{-1}$ for water peak feature, indicating the liquid-like particle appears to be amorphous solid.*

[Figure]

***Figure S3.*** *Optical micrographs of nitrate salt particles at low RH less than 10%: ammonium nitrate (upper), sodium nitrate (left), and calcium nitrate (right). The Raman spectrum is also shown for ammonium nitrate.*

***Related changes in the revised manuscript:***

*Figure S3 is added into the supplement.*

***Page 6 line 23: We add*** *"As shown in Fig. S3, the nitrate salt particles stay in a liquid-like state at low RH less than 10%. The Raman spectrum of ammonium nitrate shows no obvious absorption between 3400 cm$^{-1}$ and 3600 cm$^{-1}$ for water peak feature, indicating the liquid-like particle appears to be amorphous solid.".*

**P7L10-24: Should move these lines (about discussion of ZSR and AIOMFAC models) to section 2.**

*Reply:* *These lines about the discussion of ZSR and AIOMFAC models have been moved to section 2 in the revised manuscript.*

**P7L5-6: Malonic acid and phthalic acid exhibited continuous-water uptake in whole RH rage as the authors stated. This means the initial phase state of these acids is amorphous?**

*Reply:* *Yes, the initial phase state of these acids (malonic acid and phthalic acid) is amorphous as indicated by the continuous water uptake in the whole RH range.*

**P9L18-19: Report the growth factor value of sodium malonate here and also cite a reference about the formation of sodium acetate in ambient aerosols.**

*Reply:* *"sodium acetate" should be "sodium malonate"?* **P9L20: We add** *"Disodium malonate has a GF of 1.78 at 90% RH (Wu et al., 2011). The field measurement has observed the formation of sodium malonate in ambient aerosols (Laskin et al., 2012).".*

**P10L15-19: This statement is biased and no reasonable evidence for observed particle shrinkage. Do you have any electronic images like TEM etc.; it is hard to believe that the existence of gel-like structures in this study without showing any electronic images.**

*Reply:* *The previous HTDMA studies have shown that the specific behavior of particle shrinkage upon hydration typically results from humidity-induced transformation of porous, gel-like structures into more compact sphere for (semi-)solid amorphous particles (Mikhailov et al., 2009). As shown in Figure 6, the $Ca(NO_3)_2$/malonic acid (1:1) particle has a more compact sphere structure at 30% RH relative to the gel-like structure under dry conditions.*

[Figure]

*Figure 6. SEM micrographs of deposited particles composed of calcium nitrate and malonic acid before conditioning (left) and after conditioning at 30% RH (right).*

***Related changes in the revised manuscript:***

*Figure 6 is added into the main text.*

***Page 10 line 15: We add*** *"As shown in Fig. 6, the Ca(NO₃)₂/malonic acid (1:1) particle has a more compact sphere structure at 30% RH relative to the gel-like structure under dry conditions.".*

**The authors must and should provide more shreds of evidence for their conclusions.**

***Reply:*** *As stated above, we have added some important discussions and supplemented essential evidence for the conclusions.*

**Response to Referee #2:**

We are grateful for the reviewer's comments. Those comments are all valuable and helpful for improving our paper. Our response to the comments and changes to the manuscript are included below. We repeat the specific points raised by the reviewer in bold font, followed by our response in italic font. The pages numbers and lines mentioned below are consistent with those in the Atmospheric Chemistry and Physics Discussions (ACPD) paper.

**Jing et al. characterized the hygroscopicity of nitrate aerosols and their mixtures with several model organic acids using an HTMDA. The authors show that the presence of the organic acids can alter the phase behaviour and water uptake of nitrate aerosols, resulting in unexpected growth relative to the modelled growth. This study provides a set of valuable data for the hygroscopic behaviour of nitrate-containing aerosols generated in the lab, which has important implications for the understanding of physicochemical behaviours of atmospherically nitrate-containing particles. The data in this study appear to be of good quality. This manuscript is well organized and clearly presented. It is recommended for publication in ACP after the following minor comments are addressed.**

*Reply: We thank the reviewer for the comments. We would like to revise this manuscript according to the reviewer's comments.*

**1. P3 L14: Could the authors quantify the nitrate content in the atmospheric particulate matter?**

*Reply: We would like to add some quantitive descriptions. Chemical analyses have shown that nitrate typically constitutes a fraction (7–14%) of the total particulate matter during the high pollution events at the urban sites of China (Huang et al., 2014). The nitrate content can even dominate 22%-24% of mass fractions of particulate matter in some urban areas such as Beijing and Los Angeles (Zhang et al., 2015).*

*Related changes in the revised manuscript:*

*Page 3 line 16: We add "For example, chemical analyses have shown that nitrate typically constitutes a fraction (7–14%) of the total particulate matter during the high pollution events at the urban sites of China (Huang et al., 2014). The nitrate content can even dominate 22%-24% of mass fractions of particulate matter in some urban areas such as Beijing and Los Angeles (Zhang et al.,*

*2015).".*

**2. P6 L5: Please add a quantitative comparison to literature values.**

*Reply: Agree.*

*Related changes in the revised manuscript:*

*Page 6 line 6: We add "For example, the hygroscopic growth factors of 100 nm ammonium sulfate particles at 80% RH and 90% RH were determined to be 1.44 and 1.68, respectively, consistent with the literature results (GF(80% RH) = 1.46 and GF(90% RH) = 1.69, Sjogren et al., 2007).".*

**3. P6 L23: The authors can convert the GFs of pure nitrate particles into kappa values based on the method proposed by Petters and Kreidenweis (ACP, 2007). Such a comparison between this study and literature would likely make the results more useful to a broader audience.**

*Reply: Thanks. According to κ-Köhler theory proposed by Petters and Kreidenweis (2007), a single hygroscopicity parameter κ can be obtained based on hygroscopic growth measurements under subsaturated conditions (Carrico et al., 2008):*

$$\kappa = \frac{\left(GF^3 - 1\right)\left(1 - a_w\right)}{a_w} \tag{1}$$

*Where $a_w$ is the water activity and GF is the hygroscopic growth factor. Petters and Kreidenweis (2007) found that the κ-value derived from the hygroscopic growth data at RH~90% can be used to predict the CCN activity, and the predictions are consistent with the κ-value derived from the measured CCN data. The GFs of pure nitrate particles at 90% RH are converted into κ-values by using equation (1). The calculated κ-value in this study is 0.537, 0.821 and 0.658 for ammonium nitrate, sodium nitrate and calcium nitrate, respectively, in agreement with the corresponding value of 0.577 for ammonium nitrate and 0.80 for sodium nitrate in the literature (Petters and Kreidenweis, 2007).*

*Related changes in the revised manuscript:*

*Page 6 line 6: We add "According to κ-Köhler theory proposed by Petters and Kreidenweis (2007), a single hygroscopicity parameter κ can be obtained based on hygroscopic growth measurements under subsaturated conditions (Carrico et al., 2008):*

$$\kappa = \frac{\left(GF^3 - 1\right)\left(1 - a_w\right)}{a_w} \qquad (2)$$

*Where $a_w$ is the water activity and GF is the hygroscopic growth factor. Petters and Kreidenweis (2007) found that the κ-values derived from the hygroscopic growth data at RH∼90% are generally consistent with that derived from the measured cloud condensation nucleus (CCN) activity data. The GFs of pure nitrate particles at 90% RH are converted into κ-values by using Eq. (2).".*

***Page 6 line 23:*** *We add "The κ-values of pure nitrate particles are derived from the GFs at ∼90% RH by using Eq. (2). The calculated κ-value in this study is 0.537, 0.821 and 0.658 for ammonium nitrate, sodium nitrate and calcium nitrate, respectively, in agreement with the corresponding value of 0.577 for ammonium nitrate and 0.80 for sodium nitrate in the literature (Petters and Kreidenweis, 2007).".*

**4. P6 L27, Figure 1: It seems that AIOMFAC performs better for ammonium and sodium nitrates than calcium nitrates. Please elaborate on the possible reasons.**

***Reply:*** *The possible reason is that compared to ammonium and sodium nitrates Ca(NO₃)₂ can form amorphous hydrates under dry conditions (Gibson et al., 2006). On the basis of thermodynamic considerations, tetrahydrate (Ca(NO₃)₂·4H₂O) is the stable hydrate for calcium nitrate. The previous study has also applied tetrahydrate data to calculate the theoretical growth factors of Ca(NO₃)₂ due to the lack of detailed amorphous hydrates data (Gibson et al., 2006). Similarly, the AIOMFAC predictions for Ca(NO₃)₂ in our study are based on the tetrahydrate data (density and molar mass). It can be expected that the discrepancies between properties (density and molar mass) of amorphous hydrates and tetrahydrate could result in the slight deviation between AIOMFAC predictions and measurements.*

***Related changes in the revised manuscript:***

***Page 6, Line 29:*** *We add "Compared to ammonium and sodium nitrates, Ca(NO₃)₂ can form amorphous hydrates under dry conditions (Gibson et al., 2006). On the basis of thermodynamic considerations, tetrahydrate (Ca(NO₃)₂·4H₂O) is the stable hydrate for calcium nitrate. The previous study has also applied tetrahydrate data to calculate the theoretical growth factors of Ca(NO₃)₂ due to the lack of detailed amorphous hydrates data (Gibson et al., 2006). Similarly, the AIOMFAC predictions for Ca(NO₃)₂ in our study are based on the tetrahydrate data. It can be*

*expected that the discrepancies between properties (density and molar mass) of amorphous hydrates and tetrahydrate could result in the slight deviation between AIOMFAC predictions and measurements.".*

**5. Can the authors explain whether AIOMFAC predictions are from fitted data or from an ab initio prediction? If AIOMFAC is from a fit to data, please cite that dataset.**

*Reply: The AIOMFAC predictions for nitrate salts are based on fitted water activity data. The fitted water activity data of ammonium nitrate are derived from measurements by Hamer and Wu (1972) and Robinson and Stokes (2002). The sodium nitrate data are based on measurements from Hamer and Wu (1972) and Tang and Munkelwitz (1994). The fitted water activity data of calcium nitrate are derived from measurements by Robinson and Stokes (2002).*

*The references related to the dataset have been added in the revised manuscript.*

*Related changes in the revised manuscript:*

*Page 6, Line 29: We add "The AIOMFAC predictions for nitrate salts are based on fitted water activity data. The fitted water activity data of ammonium nitrate are derived from measurements by Hamer and Wu (1972) and Robinson and Stokes (2002). The sodium nitrate data are based on measurements from Hamer and Wu (1972) and Tang and Munkelwitz (1994). The fitted water activity data of calcium nitrate are derived from measurements by Robinson and Stokes (2002).".*

**6. P7 L6: Please quantify how your results are consistent with the observations in the literature.**

*Reply: We add some quantitive descriptions in this section.*

*Related changes in the revised manuscript:*

*Page 7 line 7: We add "The measured GF of 100 nm malonic acid particles at 80% RH and 90% RH is 1.34 and 1.65, respectively, consistent with corresponding literature value of 1.37 and 1.73 (Prenni et al., 2001). 100 nm phthalic acid particles have a GF of 1.14 at 80% RH, close to the value of ~1.12 reported by Brooks et al. (2004).".*

**7. P7 L8, section 3.2: Comparison with the literature results of ammonium sulphate is encouraged when appropriate.**

*Reply: According to reviewer's suggestion, we supplement some discussions regarding the literature*

*results of ammonium sulphate in section 3.2.*

***Related changes in the revised manuscript:***

***Page 8 line 18:*** *We add "Compared to the hygroscopic growth of $NH_4NO_3$/OA mixed particles, oxalic acid has slight influence on the deliquescence transition of ammonium sulfate, as indicated by the slightly decreased deliquescence point of mixed particles relative to pure ammonium sulfate (Jing et al., 2016; Wang et al., 2017).".*

***Page 8 line 22:*** *We add "However, the malonic acid could dramatically influence the deliquescence behavior of ammonium sulfate by promoting water uptake of mixed particles under low RH conditions (Prenni et al., 2003).".*

***Page 8 line 31:*** *We add "Compared to ammonium nitrate, the deliquescence behavior and water uptake of ammonium sulfate is almost not affected by succinic acid (Prenni et al., 2003).".*

**8. P9 L30-31: Please quantify the limited solubility. The occurrence of liquid-liquid phase separation is also influenced by the oxygen-carbon ratio of organic components. If possible, please expand the discussion regarding liquid-liquid phase separation. Thus, it can be helpful to the readers who are not familiar with this issue.**

***Reply:*** *We thank for the reviewer's valuable suggestion. It has been found that a particle containing one dicarboxylic acid with limited solubility (<1 g/100 mL) and ammonium sulfate would form phase-separated structures upon drying (Veghte et al., 2014). The prior studies observed that liquid–liquid phase separation never occurred for mixed systems containing organic species with the oxygen-to-carbon elemental ratio (O : C) ≥ 0.8 and always occurred for O:C <0.5 (You et al., 2013). Phthalic acid ($C_8H_6O_4$) has a solubility of 0.41 g/100 mL and an oxygen-carbon ratio of 0.5. Considering the low solubility and oxygen-carbon ratio of phthalic acid, it can be expected the occurrence of phase-separated structures for phthalic acid containing particles.*

***Related changes in the revised manuscript:***

***Page 10 line 1-2:*** *The sentence "Considering phthalic acid with limited solubility, the phase separation behavior may also occur for $NaNO_3$/PA mixed particles upon dehydration." **is changed into** "It should be noted that phthalic acid has the much lower solubility (0.41 g/100 mL) and oxygen-carbon ratio (O : C = 0.5) than other organic acids such as oxalic acid (9.5 g/100 mL, O : C = 2), malonic acid (76 g/100 mL, O : C = 1.33) and succinic acid (8.35 g/100 mL, O : C = 1). It*

*has been found that a particle containing one dicarboxylic acid with limited solubility (< 1 g/100 mL) and ammonium sulfate would form phase-separated structures upon drying (Veghte et al., 2014). The prior studies observed that liquid–liquid phase separation never occurred for mixed systems containing organic species with the oxygen-to-carbon elemental ratio (O : C) larger than 0.8 and always occurred for O : C less than 0.5 (You et al., 2013). Considering the low solubility and oxygen-carbon ratio of phthalic acid, it can be expected the occurrence of phase-separated structures for phthalic acid-containing particles. As shown in Fig. S2, a cover layer can be observed at the border of the dry $NaNO_3$/phthalic acid mixed particle.".*

**9. P11 L25: The findings reported in this study are likely significant for areas where nitrate concentrations are high. The authors may consider adding some specific areas where your suggested phenomena would be more pronounced.**

*Reply: Thanks. We add some specific areas with high nitrate concentrations in the revised manuscript.*

*Related changes in the revised manuscript:*

*Page 11 line 18: We add "Due to the enhanced fractions of nitrate in particulate matter and controlled sulfate by policies (Hodas et al., 2014), the nitrate may play an important role in the hygroscopicity and thus the water content of aerosol particles in urban areas such as Eastern United States and Eastern China where nitrate concentrations are high (Zhang et al., 2015).".*

**References**

[revised manuscript text omitted]

---

## Author Response (AR2)

**Author's Response**

**Response to Referee #1:**

Our response to the comments and changes to the manuscript are included below. We repeat the specific points raised by the reviewer in bold font, followed by our response in italic font. The pages numbers and lines mentioned below are consistent with those in the revised paper.

**The present manuscript investigates the hygroscopic behaviour of nitrate salts and their mixtures with organic acids using a HTDMA and model fittings. The inferences are further supported by the modelling studies and other SEM micrographs. Since nitrate aerosols are ubiquitous and well mixed with organic compounds in the atmosphere, the present study makes sense and relevant to the atmospheric aerosol. This study provides important information of growth factor data and their hygroscopic behaviours in the aqueous-phase atmosphere. The authors revealed significant information related to better understand the phase behaviour of nitrogen derived inorganic and organic salt mixtures in the atmosphere. Since hygroscopicity relate to optical and physical properties of aerosols, directly impacts the direct and indirect of earth's radiative forcing. Overall, the manuscript is well written, easy to understand and interesting too. The figures are very clear and reflect the main text. It is recommended for publication in ACP after the following minor revision as given below.**

*Reply: Thanks for the positive comments on our manuscript. We would like to revise this manuscript according to the reviewer's comments.*

**1. Page 3 Line 9-12: Atmospheric process or chemical aging also play an important role in the hygroscopic growth of aerosol particles as literature sound. In this view, recently, Professor Kawamura and his group published a couple of papers on impact of chemical composition, mixing state and chemical aging on hygroscopic growth of water-soluble aerosols in the atmosphere. I suggest the authors should discuss and cite following studies done by them here.**

**Boreddy, Kawamura et al. (2015), Long-term (2001–2012) observation of the modelled hygroscopic growth factor of remote marine TSP aerosols over the western North Pacific: impact of long-range transport of pollutants and their mixing states, Physical Chemistry Chemical Physics 17 (43), 29344-29353.**

**Boreddy, Kawamura et al. (2014), Hygroscopic behaviour of water-soluble matter extracted from Biomass burning aerosols collected at a rural site in Tanzania, East Africa. Journal of Geophysical Research-Atmospheres, 119, 12233-12245.**

*Reply: We would like to cite these studies in the manuscript.*

*Related changes in the revised manuscript:*

*Page 3 line 11: We add "Atmospheric process or chemical aging plays an important role in the chemical composition, mixing state of aerosol particles and thus impacts aerosol hygroscopicity (Boreddy et al., 2015; Boreddy et al., 2014b).".*

**2. Page 3 Line 19-21: What about the formation of KNO₃ and Mg(NO₃)₂ salts in the atmosphere? not significant?**

*Reply: Compared to NH₄NO₃, NaNO₃ and Ca(NO₃)₂ studied in our work, the formation of KNO₃ and Mg(NO₃)₂ salts in atmospheric aerosols is not significant due to the minor content of potassium and magnesium element in atmospheric particles (Zhang et al., 2015).*

**3. Page 3 Line 32: References are required here as given below which showed how WSOC affect the hygroscopicity of sea-salt particles.**

**Boreddy and Kawamura (2016), Hygroscopicity of water-soluble matter extracted from the western North Pacific aerosols: influence of atmospheric processes and long-range transport, Science of the Total Environment, 557-558, 285-295.**

**Boreddy, Kawamura et al. (2014), Hygroscopic Properties of particles nebulized from water extracts of aerosols collected at Chichijima Island in the western North Pacific: an outflow region of Asian dust. JGR- Atmos, 119, 167-178.**

*Reply: References you mentioned are included in the revised manuscript.*

*Related changes in the revised manuscript:*

*Page 3 line 32: After "which affect the hygroscopicity of inorganic components", references are included as "which affect the hygroscopicity of inorganic components (Boreddy et al., 2014a; Boreddy and Kawamura, 2016).".*

**4. Solubility also plays an important role on hygroscopicity as literature said. What is the solubility role on reported salts and their mixtures in this study? I suggest the authors should state in their manuscript if possible. It would be helpful to readers to better understand the hygroscopicity of aerosols in the atmosphere.**

*Reply: In this study, the role of solubility is largely reflected by the hygroscopic growth of oxalic or succinic acid containing aerosol particles. The oxalic and succinic acids have a limited solubility less than 10 g/100 ml and a deliquescence point larger than 97% RH. As stated in our manuscript, crystalline oxalic and succinic acids exhibit no hygroscopic growth in the RH range studied (0-90%). In contrast, the pure nitrate salts show*

*continuous water uptake in the RH range studied. As shown in Figures 2 and 4, the measured growth factors for oxalic or succinic acid containing nitrate particles are higher than that of ZSR predictions based on GF=1 (no dissolution) at high RH. Considering the dissolution of oxalic acid, the ZSR predictions based on GF of aqueous oxalic acid phase could well describe the hygroscopic growth of mixed particles at high RH, as seen in Figures 2 and 4. As stated in our manuscript, the dissolution of oxalic and succinic acids could contribute to the enhanced water uptake by mixed particles at high RH.*

*Related changes in the revised manuscript:*

[revised manuscript text omitted]